# Health-related quality of life among patients with type 2 diabetes mellitus in Eastern Province, Saudi Arabia: A cross-sectional study

Dhfer Alshayban◉, Royes Joseph◉*

Department of Pharmacy Practice, College of Clinical Pharmacy, Imam Abdulrahman Bin Faisal University, Dammam, Saudi Arabia

◉ These authors contributed equally to this work.
* rjchacko@iau.edu.sa

## Abstract

Diabetes mellitus has reached epidemic levels, and it threatens the economy and health globally and Saudi Arabia in particular. The study assessed health-related quality of life using EuroQol instrument and its predictors among patients with Type 2 diabetes mellitus in Eastern Province, Saudi Arabia. A cross-sectional study was conducted among 378 patients with Type 2 diabetes mellitus from two major health centers in Eastern Province. The study showed moderate health-related quality of life, as reported by the median index score of 0.808 with more than a quarter of patients with severe-extreme health state in some or all domains. Multiple-regression models showed that male gender, high monthly income, having no diabetes-related complications and having random blood glucose level less than 200 mg/dl were prone to have a higher index score compared to the corresponding contrary groups. The study will help in guiding the development of effective intervention programs to improve diabetes-related health-related quality of life among the Saudi population.

## Introduction

Diabetes mellitus (DM) and related complications have reached epidemic levels, and it threatens the economy and health globally. According to the International Diabetes Federation (IDF) reports, 1 in 11 adults aged 20–79 years (425 million adults; 451 million if the age is expanded to 18–99 years) had DM globally in 2017, and 90% of them were with type 2 diabetes mellitus (T2DM) [1,2]. The prevalence and incidence of DM are increasing worldwide, and a rapid progression has been reported in middle- and low-income countries [1]. The new edition of the IDF Diabetes Atlas (8th ed. 2017) reports that approximately 9.2% of adults aged 18–99 years (39.9 million people) had DM in the Middle East and North Africa Region (MENA) in 2017 [1]. It is expected that the number of people with diabetes in the MENA region will be more than double by 2045 [1]. Based on the IDF Diabetes Atlas report, Saudi Arabia is on the top among the MENA countries with the highest age-adjusted DM prevalence of 17.7%, and 4th place in terms of the number of people with diabetes [1]. IDF predicts that

**Data Availability Statement:** We have received consent from the participants for participation in the research and publication of results. However, we were not consented for sharing the data

publicly by the participants. For queries related to the study data, Dr. Mohamed Baraka, member of IRB, IAU can be contacted on his email: mabaraka@iau.edu.sa (IRB approval number: IRB-2109-05-391).

**Funding:** The authors received no specific funding for this work.

**Competing interests:** The authors have declared that no competing interests exist.

approximately one in four adults in Saudi Arabia will have diabetes by 2045 [1]. These estimates indicate that DM has reached epidemic levels, and the chronic condition threatens the global economy and health as it drains national health care budgets and reduces productivity [1,3].

DM is a significant and growing healthcare challenge in Saudi Arabia primarily because of increased physical inactivity, consumption of unhealthy diets, obesity and sedentary lifestyles [4,5]. DM is a major cause of blindness, kidney failure, heart attacks, stroke and lower limb amputation [6]. DM and its complications have contributed tremendously to the burden of mortality and disability worldwide [6]. The Global Burden of Disease Study 2015 identified DM as the ninth major cause of reduced life expectancy and reported that high fasting level of glucose was the third most common global risk factor for disability-adjusted life years in 2015 [7]. WHO reports that diabetes was the seventh leading cause of death in 2016 [3]. According to IDF Diabetes Atlas 2017, an estimated four million deaths were directly caused by DM [1]. In Saudi Arabia, the expected number of death due to DM was 14700 in 2017, and 70% of them were expected to be aged under 60 years [1].

Quality of life (QoL) indicators are solid predictors of an individual's competence to maintain long-term health, well -being and productivity [8]. Improved QoL has been regarded as a key goal of all healthcare interventions including DM management programs [9]. Previous studies reported that DM and its complications drain a substantial portion of the national healthcare budget in Saudi Arabia [1,10]. Hence, it is important to know the level of health-related QoL (HRQoL) of diabetes patients against the huge spending from the national budget. Identifying factors that are associated with impaired HRQoL may help policymakers to prioritize funding and implement interventions to improve the QoL.

Studies from Saudi Arabia [11–13], other Middle Eastern countries [14–16], and rest of the world [17,18] show that diabetes impairs the QoL of patients, but the level of impairment was not the same across the studies. A recent review indicated that Saudi Arabia's direct spending on diabetes was almost 14% of the total health expenditure, and the study urged for improving health and HRQoL of diabetes patients in order to reduce the social and personal costs for diabetes care in Saudi Arabia [5]. Apart from three regional level studies (from Makkah and Riyadh regions), a national level study on HRQoL among diabetes patients in Saudi Arabia has not been reported during the past decade. Importantly, any QoL studies among diabetes patients from Eastern Province, the largest region of Saudi Arabia, has not been reported previously. EQ-5D is regarded as one of generic instruments, rather than a disease-specific, that has been used extensively in research recently beside other instruments such as SF-36 [19,20]. Among these instruments, the EQ-5D has the benefit of being able to convert health states into a single index value that can be compared among diseases and used for economic evaluation [21]. Therefore, the present study used the EQ-5D instrument to measure HRQoL in T2DM patients in Eastern Province, Saudi Arabia, and to determine the impact of socio-demographic and clinical factors on HRQoL.

## Methodology

### Study setting and subjects

A cross-sectional study was conducted from November 2017 to April 2018 among 378 T2DM patients. Patients were conveniently recruited from two health centers of the King Fahad Hospital of the University, which is a major tertiary hospital in the Eastern Province, Saudi Arabia. One center is located in the Khobar and Dhahran region, and the other one is located in the Dammam region. Hospital statistics of these health centers and collected demographic data of

patients indicated that fair representation of patients from several geographical locations within the Eastern Province. A minimum sample size of 385 was calculated by assuming 50% of patients were adherent to treatments with the absolute precision of 0.05 and 95% confidence level. The 50% was purposively selected so that it provided the largest minimum sample size. Patients with minimum age of 18 years and with T2DM for at least 1 year were considered for this study if they provided a written informed consent. Patients with pregnancy or other medical complications were excluded from the study. The study was approved by the Institutional Review Board and the Ethical Committee at Imam Abdulrahman Bin Faisal University (IRB-2019-05-391).

## Data collection

An Arabic version of the EQ-5D questionnaire was used after obtaining prior permission from the EuroQol Research Foundation [22,23]. The participants were interviewed in Arabic, and their socio-demographic and clinical characteristics were obtained. The EQ-5D questionnaire was filled by the participants.

**Socio-demographic and clinical characteristics.** The data on participants' gender, age, education status, monthly income, number of diabetes-related complications, current use of anti-diabetic medications (type and number), and random blood glucose level were collected.

**Assessment of HRQoL.** HRQoL was assessed using the EQ-5D-5L [24]. The EQ-5D-5L involves patient self-reporting of their health status in terms of five dimensions: mobility, self-care, usual activities, pain/discomfort, and anxiety/depression. Each dimension has a five-level severity scale (no problems, slight, moderate, severe and extreme) scored from 1 to 5. Five-digit codes for the HRQoL of each patient are obtained from the score digits; there are 3125 possible sets of values, called health states, for EQ-5D-5L. The health states would range from 11111 (perfect health) to 55555 (worst health) and can be converted into a single weighted index score (EQ-5D index) using population preference scores. We used the EQ-5D-5L value set for England to derive the EQ-5D index [21]. Thus, a health state yields index score of between -0.285 and 1: index score 1 represents perfect health, 0 represents a health state equivalent to death and a score of less than 0 represents health states worse than death.

## Statistical analysis

Socio-demographic and disease characteristics of the participants were summarized using descriptive statistics. Percentages and frequencies were used for the categorical variables, while median and interquartile range were calculated for the continuous variables. Association of socio-demographic and clinical factors on HRQoL was assessed using three approaches: 1). Using chi-square test where EQ-5D health states were divided into three categories (perfect health indicates no problem in domains of EQ-5D; slight/moderate indicates problems in some domains but not worse than moderate health in any domains; severe/unable indicates a health status with problems worse than moderate health in some domains). 2). Using a multiple logistic regression with forward selection (likelihood ratio) of predictor variables, where the outcome variable was a binary variable indicating 'perfect health' (EQ-5D index = 1.000) or 'imperfect health' (EQ-5D index <1.000). 3). Using a multiple linear regression where the dependent variable was the cubic function of EQ-5D index score (the cubic function ensured normally distributed residuals). A p-value less than 0.05 was considered as statistically significant. All analyses were carried out using SPSS Statistics 24.0.

## Results

### Socio-demographic and clinical characteristics of participants

Table 1 presents the socio-demographic and clinical characteristics of participants. Among the 378 participants, half of them were male, 79% were older than 50 years, more than 50% had an education of high school or more, and more than half of them had monthly income of 5000sar or more. Regarding the clinical characteristics of the participants, 78% had diabetes related complications, 61% were on oral anti-diabetic medications only and 70% were on multiple anti-diabetic medications. 48% of participants had random glucose level of 200 mg/dl or more.

### Health-related quality of life

Fig 1 shows the patients' response over five levels in each of the five domains of EQ-5D. Among the respondents, 88%, 51%, 50%, 43% and 31% were agreed as having no problem in terms of self-care, anxiety or depression, usual activities, mobility, and pain or discomfort

**Table 1. Socio-demographic and clinical characteristics of participants (N = 378).**

| Variables | n (%[#]) |
|---|---|
| Gender | |
| Male | 182 (48.1%) |
| Female | 186 (49.2%) |
| Age | |
| <50 years | 78 (20.6%) |
| >50 years | 298 (78.8%) |
| Education status | |
| Primary or lower | 176 (46.6%) |
| High school/Secondary | 128 (33.9%) |
| College graduate | 74 (19.6%) |
| Monthly income (in SAR) | |
| Less than 5000 | 162 (42.9%) |
| 5000 to 10000 | 98 (25.9%) |
| More than 10000 | 112 (29.6%) |
| Number of diabetes related complications | |
| Nil | 84 (22.2%) |
| One | 120 (31.7%) |
| More than one | 174 (46.0%) |
| Type of anti-diabetic medication | |
| Insulin injection or combination | 144 (38.1%) |
| Oral medication only | 230 (60.8%) |
| Number of anti-diabetic medications using | |
| One medication | 114 (30.2%) |
| Two medications | 166 (43.9%) |
| Three or more medications | 82 (21.7%) |
| Random blood glucose level | |
| less than 200 | 196 (51.9%) |
| 200 to 299 | 134 (35.4%) |
| More than 300 | 48 (12.7%) |

[#]few observations were missing on some variables, but % was calculated based on 378.

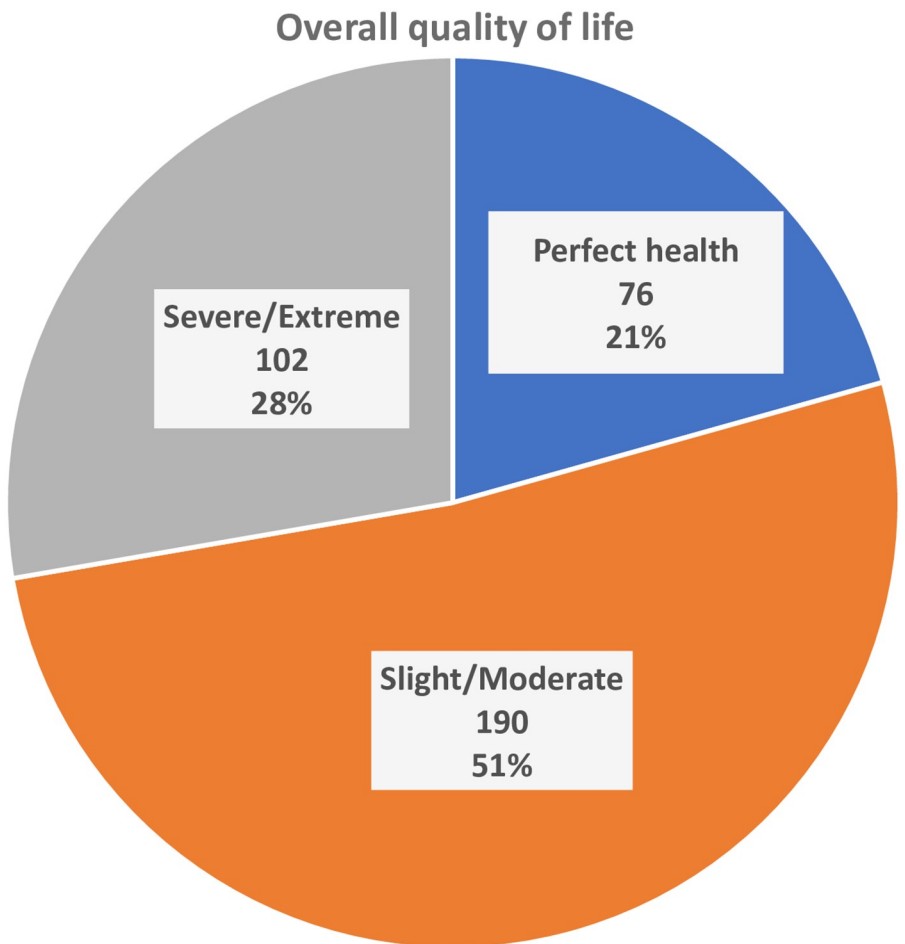

**Fig 1. Health-related quality of life measured using EQ-5D-5L scale.**

respectively. In combined, as shown in Fig 2, one-fifth (76/368) of patients did not have a problem in any domains of EQ-5D (called perfect health state); half of the participants (190/368) reported as having problems in some domains but not worse than moderate health in any domains (called slight-moderate health state); and the remaining 28% (162/368) reported as having problems worse than moderate health in some domains (called severe-extreme health state). The median (interquartile range) EQ-5D index was 0.808 (0.647–0.937).

Table 2 presents the association of socio-demographic and clinical factors on the level of HRQoL based on univariate analyses. It reports the percentage of participants with perfect health, slight-moderate health and severe-moderate health states within each level of factors of interest. A higher percentage of participants with perfect health state and a lower percentage of participants with severe-extreme health state were reported among male gender (p-value<0.001), patients aged 50 years or lower (p-value = 0.026), college graduates (p-value<0.001), patients with monthly income greater than 5000sar (p-value<0.001), patients having no diabetes-related complications (p-value<0.001), patients taking only oral anti-diabetic medication (p-value = 0.014), and patients with RBG less than 200 mg/dl (p-value<0.001) compared to that in the corresponding contrary groups.

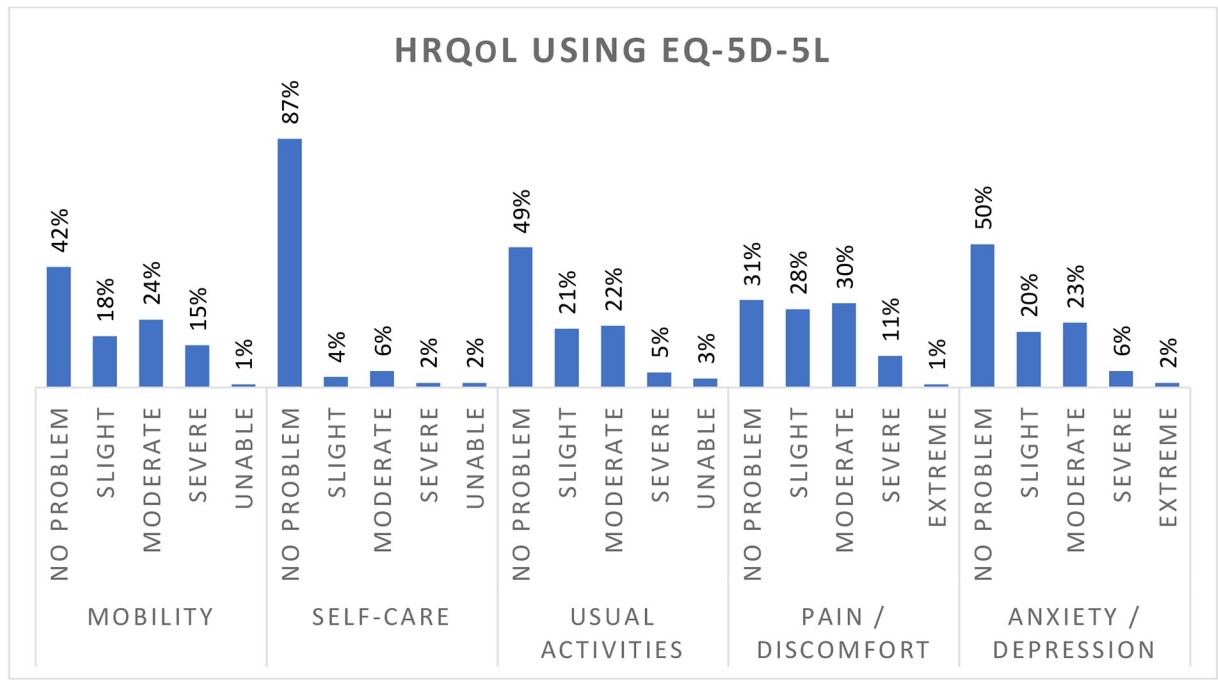

**Fig 2. Overall health-related quality of life.** Perfect health indicates no problem in domains of EQ-5D; Slight/moderate indicates problems in some domains but not worse than moderate health in any domains; Severe/unable indicates a health status with problems worse than moderate health in some domains. 10 participants did not respond to some domains.

### Predictors of severe/extreme health state: Binomial modeling

Results from a multiple logistic regression, where the outcome variable was a binary variable indicating 'severe/extreme health state in some or all domain' or 'not', is presented in Table 3. Table 3 reports the adjusted odds ratio and its 95% confidence interval for the considered factors. The adjusted odds ratio for gender indicates that the odds of having severe/extreme health state among females was nearly six-fold of that among males (p-value<0.001). Similarly, the odds of having severe/extreme health state among patients with RBG >300 mg/dl and RBG in between 200–300 mg/dl were nearly threefold (p-value = 0.001) and twofold (p-value = 0.076), respectively, compared to that among patients with RBG<200 mg/dl. In addition, patients with more than one diabetes-related complications (adjusted OR = 3.5) and patients with monthly income between 5000 to 10000 (adjusted OR = 0.13) also showed a significant association with the health states.

### Predictors of EQ-5D index score: Continuous modeling

Results from a multiple linear regression, where the dependent variable was the cubic function of EQ-5D index score, is presented in Table 4. Predictor variables in the logistic regression model were included. Table 4 reports the adjusted regression coefficient and its 95% confidence interval. The results confirm the findings from the logistic model that male gender, monthly income greater than 5000 SAR, having no diabetes-related complications and having RBG less than 200 mg/dl were prone to have a higher EQ-5D index compared to the corresponding contrary groups.

**Table 2. Overall health related quality of life (Univariate analysis).**

| Factors | Overall health status | | | |
|---|---|---|---|---|
| | **Perfect health** | **Slight/Moderate** | **Severe/Extreme** | **p-value[a]** |
| Gender | | | | |
| Male | 64 (35.6%) | 92 (51.1%) | 24 (13.3%) | <0.001 |
| Female | 12 (6.7%) | 94 (52.8%) | 72 (40.4%) | |
| Age | | | | |
| 50 years | 14 (18.9%) | 48 (64.9%) | 12 (16.2%) | 0.026 |
| 50 years | 62 (21.2%) | 142 (48.6%) | 88 (30.1%) | |
| Education status | | | | |
| Primary or lower | 24 (14.1%) | 78 (45.9%) | 68 (40%) | <0.001 |
| High/Secondary | 26 (20.6%) | 76 (60.3%) | 24 (19%) | |
| College graduate | 26 (36.1%) | 36 (50%) | 10 (13.9%) | |
| Monthly income (in SAR) | | | | |
| Less than 5000 | 18 (11.4%) | 74 (46.8%) | 66 (41.8%) | <0.001 |
| 5000 to 10000 | 26 (27.7%) | 60 (63.8%) | 8 (8.5%) | |
| More than 10000 | 32 (29.1%) | 54 (49.1%) | 24 (21.8%) | |
| Number of diabetes related complications | | | | |
| Nil | 24 (29.3%) | 48 (58.5%) | 10 (12.2%) | <0.001 |
| One | 36 (30%) | 58 (48.3%) | 26 (21.7%) | |
| More than one | 16 (9.6%) | 84 (50.6%) | 66 (39.8%) | |
| Type of anti-diabetic medication | | | | |
| Insulin injection or combination | 20 (14.1%) | 74 (52.1%) | 48 (33.8%) | 0.014 |
| Only oral medication | 56 (25.2%) | 114 (51.4%) | 52 (23.4%) | |
| Number of anti-diabetic medications using | | | | |
| One medication | 26 (23.6%) | 64 (58.2%) | 20 (18.2%) | 0.101 |
| Two medications | 34 (21.3%) | 76 (47.5%) | 50 (31.3%) | |
| Three or more | 16 (19.5%) | 38 (46.3%) | 28 (34.1%) | |
| Random blood glucose level | | | | |
| less than 200 | 64 (33.7%) | 94 (49.5%) | 32 (16.8%) | <0.001 |
| 200 to 299 | 10 (7.6%) | 72 (54.5%) | 50 (37.9%) | |
| More than 300 | 2 (4.3%) | 24 (52.2%) | 20 (43.5%) | |

[a]Chi-square test was used

## Discussion

The burden of T2DM in Saudi Arabia is steadily increasing due to population growth, urbanization, lack of physical activity and unhealthy diet [4,25,26]. HRQoL is one of the important outcomes used to evaluate the effect of management of chronic diseases on health, and it reflects a patient's physical and psychosocial disease burden. The present study used EQ-5D-5L to measure the HRQoL for the first time in the Arab region. Previous studies support the use of EQ-5D-5L over EQ-5D-3L as the scale with five levels has more discriminative power than the scale with three levels in patients with T2DM [27]. The present study showed moderate HRQoL with the median EQ-5D index score of 0.808. A similar finding was reported by an earlier study conducted in the Riyadh region, Saudi Arabia with a mean EQ-5D index of 0.70 [11,28]. The difference in the index score may be due to the choice of the number of levels in EQ-5D, the selection of participants to study, the quality of diabetes care, or the availability of access to support services. Another two studies conducted in Riyadh and Makah regions, Saudi Arabia, but used different measurements scales, also affirmed our finding [12,13]. A

**Table 3. Predictors of severe/extreme health status—Adjusted odds ratio (AOR) and its 95% confidence interval.**

| Factors | Odds ratio (95% CI) | p-value |
|---|---|---|
| Gender | | |
| Male | Reference | |
| Female | 5.58 (2.78–11.2) | <0.001 |
| Monthly income (in SAR) | | |
| Less than 5000 | 1.80 (0.89–3.64) | 0.104 |
| 5000 to 10000 | 0.13 (0.04–0.42) | 0.001 |
| More than 10000 | Reference | |
| Number of diabetes related complications | | |
| Nil | Reference | |
| One | 2.24 (0.78–6.45) | 0.136 |
| More than one | 3.54 (1.32–9.50) | 0.012 |
| Type of anti-diabetic medication | | |
| Oral medication only | Reference | |
| Insulin injection or combination | 1.12 (0.46–2.72) | 0.806 |
| Random blood glucose level | | |
| less than 200 | Reference | |
| 200 to 299 | 3.05 (1.55–6.00) | 0.001 |
| More than 300 | 2.18 (0.92–5.13) | 0.076 |

recent study from Jordan reported a similar mean EQ-5D index of 0.724 in T2DM patients in Jordan [14]. Our estimate is also consistent with findings from neighboring Middle Eastern countries [15,16]. Even though our study identified an overall moderate HRQoL, 79% of patients still had imperfect health state on some EQ-5D domains and 28% of patients reported a severe-extreme health state. Specifically, only 31% and 43% of patients expressed no problem in terms of pain/discomfort and mobility respectively. Hence, it is important to assess the influencing factors of HRQoL in patients with T2DM for the better planning of interventions to improve the physical and psychosocial burden of the disease, and hence to attain better HRQoL.

**Table 4. Summary of multiple linear regression model for cubic function of EQ-5D index.**

| Factors | Estimate (95% CI) | p-value |
|---|---|---|
| Gender | | |
| Male | Reference | |
| Female | -0.19 (-0.24, -0.13) | <0.001 |
| Monthly income (in SAR) | | |
| Less than 5000 | Reference | |
| 5000 to 10000 | 0.10 (0.03, 0.16) | 0.004 |
| More than 10000 | 0.17 (0.1, 0.24) | <0.001 |
| Number of diabetes related complications | | |
| Nil | Reference | |
| One | -0.20 (-0.27, -0.12) | <0.001 |
| More than one | -0.10 (-0.17, -0.02) | 0.013 |
| Random blood glucose level | | |
| less than 200 | Reference | |
| 200 to 299 | -0.24 (-0.33, -0.15) | <0.001 |
| More than 300 | -0.18 (-0.24, -0.12) | <0.001 |

Previous studies have reported a lower HRQoL among female with diabetes compared to male with diabetes [11,12,14,29–33]. The present study also reported that HRQoL is gendered in favor of male patients with T2DM. The multivariate analysis indicated female gender as an independent predictor of poor HRQoL. The adjusted odds ratio indicates that the odds of having severe/extreme health state among females was 5.5 times higher than that among males. The multiple linear regression also confirms a higher EQ-5D index score among male patients compared to female patients. A recent systematic review reported a substantial difference in the level of physical activity favoring men in the Arab countries [34]. In addition, the socio-cultural differences between men and women in the Arab world could be a reason for the gender difference in the HRQoL. Therefore, identifying strategies to improve the quality of life among patients with diabetes, especially among women, is of great importance.

Aging has been identified as a key factor for T2DM [4,25,26] and impaired HRQoL [12,15,35,36]. Therefore, it is expected a negative association between age and HRQoL among patients with diabetes. In the present study, 30% of older patients with T2DM reported severe-extreme impaired HRQoL compared to 16% among patients aged less than 50 years. The result was not statistically significant in the regression models, which may be due to the fewer representation of younger patients in our study sample.

Studies have demonstrated that socioeconomic status is positively associated with HRQoL among adults with a chronic disease [37]. In the present study, a higher proportion (40%) of patients having primary education or lower reported severe-extreme impaired HRQoL compared to patients having higher education; which is consistent with a previous study from Oman [15]. Similarly, a higher proportion (42%) of patients with low monthly income reported severe-extreme impaired HRQoL compared to patients having moderate/high monthly income. However, the multiple regression models did not find a significant difference in EQ-5D index between the educational levels. As Robert et al pointed out in a study, an improvement in HRQoL of people at the lowest end of the socioeconomic distribution helps substantial improvement in the HRQoL at the population level [38].

The present study reports a higher proportion, but not statistically significant in the logistic regression model, of patients with severe-extreme impaired HRQoL among patients under insulin therapy compared to the contrary group. The difference in EQ-5D index score was also not significant based on the multiple regression model. HRQoL can be positively and negatively associated with insulin therapy [39]. Due to the beneficial effects of the insulin therapy, such as better glycemic control and lower risk of diabetic complications, better rating may have been given on domains of HRQoL [40]. Conversely, the inconvenience associated with the insulin therapy, fear of weight gain and the risk of hypoglycemia may adversely affect the patient's HRQoL [40]. In the current study, having multiple complications of diabetes was found to be negatively associated with HRQoL. Patients who had more than one complications reported lower EQ-5D score, in consistent with other studies that explored the relationship [12].

Many studies have previously reported that the severity of T2DM has a negative impact on quality of life [41,42], however, the impact of the level of RBG on HRQoL is still uncertain. Kayo et al reported RBG level was negatively associated with cognitive impairment in the elderly [43]. A recent study reported a non-significant relationship between quality of life with glucose levels among Iranian diabetic patients [44]. However, our study showed a strong negative association between random glucose level and HRQoL. The adjusted odds ratios indicate that the odds of having severe-extreme health state among patients with RBG >200 mg/dl was more than twofold of that among patients with RBG<200 mg/dl. In addition, the multiple linear regression also confirmed a significant reduction in EQ-5D index score against an increase

in RBG level. A high RBG usually account for the poor control of diabetes, and hence it may negatively affect the HRQoL.

Some limitations should be noted. The study was of a cross-sectional design and thus the association that has been demonstrated in our study may not imply a causal relationship. Importantly, the study was restricted to patients from two outpatient health centers of a major tertiary hospital in Eastern Province, Saudi Arabia. However, a study on diabetes-related QoL has not been reported at the national level or from Eastern Province in the past decade, and hence the importance of our study. Although the study was restricted to patients with a minimum one-year duration of diabetes, the actual duration was not obtained. HbA1c, which may be a better indicator for glucose control than random glucose level, was also not collected.

## Conclusion

This study demonstrates a moderate HRQoL among patients with T2DM. The impaired HRQoL is mainly in terms of pain/discomfort and mobility due to diabetes. The results showed that male gender, high income, without complications and good glucose control have relatively better quality of life. The study will help in guiding the development of effective intervention programs to improve T2DM related HRQoL among the Saudi population. Such programs should target especially at groups with female gender, older age, low socio-economic status, multiple complications of diabetes and high RBG.

## Acknowledgments

We thank Dr. Muhammed Aftab for his valuable advice on study design and selection of the HRQoL instrument. We also thank Ms. Razan Abuaziz and Ms. Zahra Alhalami for their support with the data collection.

## Author Contributions

**Conceptualization:** Dhfer Alshayban, Royes Joseph.

**Data curation:** Dhfer Alshayban.

**Formal analysis:** Royes Joseph.

**Methodology:** Dhfer Alshayban, Royes Joseph.

**Project administration:** Dhfer Alshayban.

**Supervision:** Dhfer Alshayban.

**Validation:** Dhfer Alshayban, Royes Joseph.

**Writing – original draft:** Dhfer Alshayban, Royes Joseph.

**Writing – review & editing:** Dhfer Alshayban, Royes Joseph.

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
