## [Decision Letter · Decision Letter 0]

11 Dec 2019

PONE-D-19-31014

Deterioration in health-related quality of life among patients with type 2 diabetes mellitus in Eastern Province, Saudi Arabia: a cross-sectional study

PLOS ONE

Dear Dr Joseph,

Thank you for submitting your manuscript to PLOS ONE. After careful consideration, we feel that it has merit but does not fully meet PLOS ONE’s publication criteria as it currently stands. Therefore, we invite you to submit a revised version of the manuscript that addresses the points raised during the review process.

We would appreciate receiving your revised manuscript by Jan 25 2020 11:59PM. To enhance the reproducibility of your results, we recommend that if <gwmw class="ginger-module-highlighter-mistake-type-3" id="gwmw-15759998731778913492731">applicable you</gwmw> deposit your laboratory protocols in protocols.io, where a protocol can be assigned its own identifier (DOI) such that it can be cited independently in the future. For <gwmw class="ginger-module-highlighter-mistake-type-3" id="gwmw-15759998736544486412313">instructions see</gwmw>: http://journals.plos.org/plosone/s/submission-guidelines#loc-laboratory-protocols

A rebuttal letter that responds to each point raised by the academic editor and reviewer(s). This letter should be uploaded as <gwmw class="ginger-module-highlighter-mistake-type-3" id="gwmw-15759998752790744575798">separate file</gwmw> and labeled 'Response to Reviewers'.A marked-up copy of your manuscript that highlights changes made to the original version. This file should be uploaded as <gwmw class="ginger-module-highlighter-mistake-type-3" id="gwmw-15759998767435887926120">separate file</gwmw> and labeled 'Revised Manuscript with Track Changes'.An unmarked version of your revised paper without <gwmw class="ginger-module-highlighter-mistake-type-3" id="gwmw-15759998773352251555693">tracked</gwmw> changes. This file should be uploaded as <gwmw class="ginger-module-highlighter-mistake-type-3" id="gwmw-15759998780290879901437">separate file</gwmw> and labeled 'Manuscript'.

Please <gwmw class="ginger-module-highlighter-mistake-type-3" id="gwmw-15759998791239649151156">note while</gwmw> forming your response, if your article is accepted, you may have the opportunity to make the peer review history publicly available. The record will include editor decision letters (with reviews) and your responses to reviewer comments. If eligible, we will contact you to opt in or out.

We look forward to receiving your revised manuscript.

Kind regards,

Manal S. <gwmw class="ginger-module-highlighter-mistake-type-1" id="gwmw-15759998822522493615562">Fawzy</gwmw>, Ph.D., M.D.

Academic Editor

PLOS ONE

Journal Requirements:

**When submitting your revision, we need you to address these additional requirements:**

**Please ensure that your manuscript meets PLOS ONE's style requirements, including those for file naming. The PLOS ONE style templates can be found at http://www.plosone.org/attachments/PLOSOne_formatting_sample_main_body.pdf and http://www.plosone.org/attachments/PLOSOne_formatting_sample_title_authors_affiliations.pdf**Please refer to any sample size calculation performed prior to participant recruitment. If this was not performed please justify the reasons. Please refer to our statistical reporting guidelines for assistance (https://journals.plos.org/plosone/s/submission-guidelines.#loc-statistical-reporting). Your ethics statement must appear in the Methods section of your manuscript. If your ethics statement is written in any section besides the Methods, please move it to the Methods section and delete it from any other section. Please also ensure that your ethics statement is included in your manuscript, as the ethics section of your online submission will not be published alongside your manuscript. We note that you have indicated that data from this study are available upon request. PLOS only allows data to be available upon request if there are legal or ethical restrictions on sharing data publicly. For information on unacceptable data access restrictions, please see http://journals.plos.org/plosone/s/data-availability#loc-unacceptable-data-access-restrictions.In your revised cover letter, please address the following prompts:a) If there are ethical or legal restrictions on sharing a de-identified data set, please explain them in detail (e.g., data contain potentially identifying or sensitive patient information) and who has imposed them (e.g., an ethics committee). Please also provide contact information for a data access committee, ethics committee, or other institutional body to which data requests may be sent.b) If there are no restrictions, please upload the minimal anonymized data set necessary to replicate your study findings as either Supporting Information files or to a stable, public repository and provide us with the relevant URLs, DOIs, or accession numbers. Please see http://www.bmj.com/content/340/bmj.c181.long for guidelines on how to de-identify and prepare clinical data for publication. For a list of acceptable repositories, please see http://journals.plos.org/plosone/s/data-availability#loc-recommended-repositories.We will update your Data Availability statement on your behalf to reflect the information you provide.

Reviewers' comments:

Reviewer's Responses to Questions

**Comments to the Author**

1. Is the manuscript technically sound, and do the data support the conclusions?

Reviewer #1: Yes

Reviewer #2: Yes

2. Has the statistical analysis been performed appropriately and rigorously? 

Reviewer #1: Yes

Reviewer #2: Yes

3. Have the authors made all data underlying the findings in their manuscript fully available?

The PLOS Data policy requires authors to make all data underlying the findings described in their manuscript fully available without restriction, with rare exception (please refer to the Data Availability Statement in the manuscript PDF file). The data should be provided as part of the manuscript or its supporting information, or deposited <gwmw class="ginger-module-highlighter-mistake-type-3" id="gwmw-15759998924967723228374">to</gwmw> a public repository. For example, in addition to summary statistics, the data points behind means, medians and variance measures should be available. If there are restrictions on publicly sharing data—e<gwmw class="ginger-module-highlighter-mistake-type-3" id="gwmw-15759998941497671118082">.</gwmw>g. <gwmw class="ginger-module-highlighter-mistake-type-1" id="gwmw-15759998947418434800189">participant</gwmw> privacy or use of data from a third party—those must be specified.

Reviewer #1: Yes

Reviewer #2: Yes

4. Is the manuscript presented in an intelligible fashion and written in standard English?

PLOS ONE does not copyedit accepted manuscripts, so the language in <gwmw class="ginger-module-highlighter-mistake-type-3" id="gwmw-15759998964932130596332">submitted</gwmw> articles must be clear, correct, and unambiguous. Any typographical or grammatical errors should be corrected at revision, so please note any specific errors here.

Reviewer #1: Yes

Reviewer #2: Yes

5. Review Comments to the Author

Please use the space provided to explain your answers to the questions above. You may also include additional comments for the author, including concerns about dual publication, research ethics, or publication ethics. <gwmw class="ginger-module-highlighter-mistake-type-3" id="gwmw-15759998998023000172361">(</gwmw>Please upload your review as an attachment if it exceeds 20,000 characters)

Reviewer #1: 1. In the title the word deterioration gives you the sense that you follow healthy participants and then you study deterioration for a period of <gwmw class="ginger-module-highlighter-mistake-type-3" id="gwmw-15759999025427521552251">time but</gwmw> in this study the main objective is to assess the quality of life and to look for predictors, so my suggestion is to change the title to suit with the objectives.

2. Usually the use of abbreviations is not suitable in the abstract unless you clarify them even international abbreviations.

3. The methodology needs to be revised; you have to mention how you determine the amount of sample and why you select two centers and how many centers in total in the area, all these things should be mentioned.

Reviewer #2: According to my knowledge, it is a novel paper in its field opening new horizons for further evidence. In addition, the object as well as the results are appropriately discussed in the context of previous literature explaining the importance of the manuscript in its field. <gwmw class="ginger-module-highlighter-mistake-type-3" id="gwmw-15759999074429494236327">Authors</gwmw> succeed to present their data in a clear <gwmw class="ginger-module-highlighter-mistake-type-3" id="gwmw-15759999074422992800253">way adding</gwmw> information to the existing literature.

Therefore, I have no corrections or further work to propose for the improvement of the manuscript and therefore it can be published unaltered.

6. PLOS authors have the option to publish the peer review history of their article (what does this mean?). If published, this will include your full peer review and any attached files.

If you choose “no”, your identity will remain <gwmw class="ginger-module-highlighter-mistake-type-3" id="gwmw-15759999105508331080907">anonymous but</gwmw> your review may still be made public.

Reviewer #1: No

Reviewer #2: Yes: Athanasia Papazafiropoulou

<gdiv></gdiv>

---

## [Author Response · Author response to Decision Letter 0]

19 Dec 2019

Response to academic editor

The manuscript has been modified according to the journal’s style requirement (Refer page 1, L2-3).

2. Please refer to any sample size calculation performed prior to participant recruitment. If this was not performed please justify the reasons. Please refer to our statistical reporting guidelines for assistance (https://journals.plos.org/plosone/s/submission-guidelines.#loc-statistical-reporting).

Details on sample size calculation has been added to the methods section (Refer P 5; L90-93). “A minimum sample size of 385 was calculated by assuming 50% of patients were adherent to treatments with the absolute precision of 0.05 and 95% confidence level. The 50% was purposively selected so that it provided the largest minimum sample size.” 

3. Your ethics statement must appear in the Methods section of your manuscript. If your ethics statement is written in any section besides the Methods, please move it to the Methods section and delete it from any other section. Please also ensure that your ethics statement is included in your manuscript, as the ethics section of your online submission will not be published alongside your manuscript.

Ethics statement was included in the methods section (Page 5, L100-101).

4. We note that you have indicated that data from this study are available upon request. PLOS only allows data to be available upon request if there are legal or ethical restrictions on sharing data publicly. For information on unacceptable data access restrictions, please see http://journals.plos.org/plosone/s/data-availability#loc-unacceptable-data-access-restrictions. In your revised cover letter, please address the following prompts:

 We have addressed the following in the revised cover letter.

“We have received consent from the participants for participation in the research and publication of results. However, we were not consented for sharing the data publicly by the participants. For queries related to the study data, Dr. Mohamed Baraka, member of IRB, IAU can be contacted on his email: mabaraka@iau.edu.sa (IRB approval number: IRB-2109-05-391).”

Response to reviewers:

Reviewer #1:

1. In the title the word deterioration gives you the sense that you follow healthy participants and then you study deterioration for a period of time but in this study the main objective is to assess the quality of life and to look for predictors, so my suggestion is to change the title to suit with the objectives.

We thank the reviewer for the valid point. We have modified the title accordingly. The title is “Health-related quality of life among patients with type 2 diabetes mellitus in Eastern Province, Saudi Arabia: a cross-sectional study”

2. Usually the use of abbreviations is not suitable in the abstract unless you clarify them even international abbreviations.

Corrected (Page 2)

3. The methodology needs to be revised; you have to mention how you determine the amount of sample and why you select two centers and how many centers in total in the area, all these things should be mentioned.

We have revised the methodology section by including details of sample size calculation (Page 5, L94-97) and sampling methodology (Page 5, L89-94). 

We calculated a minimum sample size of 385 assuming 50% of patients were adherent to treatments with the absolute precision of 0.05 and 95% confidence level. The 50% was purposively selected so that it provided the largest minimum sample size. As mentioned in the methodology section, we considered two health centres in the Eastern Province, Saudi Arabia out of five state owned general hospitals in the province. These two centres, which are affiliated with a largest university in the region, serves larger volume of patients from different geographical locations in the region compared to other hospitals. 

Reviewer #2: 

1. According to my knowledge, it is a novel paper in its field opening new horizons for further evidence. In addition, the object as well as the results are appropriately discussed in the context of previous literature explaining the importance of the manuscript in its field. Authors succeed to present their data in a clear way adding information to the existing literature.

Therefore, I have no corrections or further work to propose for the improvement of the manuscript and therefore it can be published unaltered.

We thank the reviewer for his good feedback.

---

## [Decision Letter · Decision Letter 1]

23 Dec 2019

Health-related quality of life among patients with type 2 diabetes mellitus in Eastern Province, Saudi Arabia: a cross-sectional study

PONE-D-19-31014R1

Dear Dr. Joseph,

We are pleased to inform you that your manuscript has been judged scientifically suitable for publication and will be formally accepted for publication once it complies with all outstanding technical requirements.

Shortly after the formal acceptance letter is sent, an invoice for payment will follow. To ensure an efficient production and billing process, please log <gwmw class="ginger-module-highlighter-mistake-type-1" id="gwmw-15768756868023048790062">into</gwmw> Editorial Manager at https://www.editorialmanager.com/pone/, click the "Update My Information" link at the top of the page, and update your user information. If you have any billing related questions, please contact our Author Billing department directly at authorbilling@plos.org.

With kind regards,

Manal S. <gwmw class="ginger-module-highlighter-mistake-type-1" id="gwmw-15768756915927293282917">Fawzy</gwmw>, Ph.D., M.D.

Academic Editor

PLOS ONE

Additional Editor Comments (optional):

The authors have adequately addressed the concerns raised by the reviewer. Thank you

Reviewers' comments:

Reviewer's Responses to Questions

**Comments to the Author**

1. If the authors have adequately addressed your comments raised in a previous round of review and you feel that this manuscript is now acceptable for publication, you may indicate that here to bypass the “Comments to the Author” section, enter your conflict of interest statement in the “Confidential to Editor” section, and submit your "Accept" recommendation.

Reviewer #1: All comments have been addressed

2. Is the manuscript technically sound, and do the data support the conclusions?

Reviewer #1: Yes

3. Has the statistical analysis been performed appropriately and rigorously? 

Reviewer #1: Yes

4. Have the authors made all data underlying the findings in their manuscript fully available?

The PLOS Data policy requires authors to make all data underlying the findings described in their manuscript fully available without restriction, with rare exception (please refer to the Data Availability Statement in the manuscript PDF file). The data should be provided as part of the manuscript or its supporting information, or deposited <gwmw class="ginger-module-highlighter-mistake-type-3" id="gwmw-15768757033404276918669">to</gwmw> a public repository. For example, in addition to summary statistics, the data points behind means, medians and variance measures should be available. If there are restrictions on publicly sharing data—e<gwmw class="ginger-module-highlighter-mistake-type-3" id="gwmw-15768757049859816216562">.</gwmw>g. <gwmw class="ginger-module-highlighter-mistake-type-1" id="gwmw-15768757055533336214388">participant</gwmw> privacy or use of data from a third party—those must be specified.

Reviewer #1: Yes

5. Is the manuscript presented in an intelligible fashion and written in standard English?

PLOS ONE does not copyedit accepted manuscripts, so the language in <gwmw class="ginger-module-highlighter-mistake-type-3" id="gwmw-15768757071859078001825">submitted</gwmw> articles must be clear, correct, and unambiguous. Any typographical or grammatical errors should be corrected at revision, so please note any specific errors here.

Reviewer #1: Yes

6. Review Comments to the Author

Please use the space provided to explain your answers to the questions above. You may also include additional comments for the author, including concerns about dual publication, research ethics, or publication ethics. <gwmw class="ginger-module-highlighter-mistake-type-3" id="gwmw-15768757103221064781348">(</gwmw>Please upload your review as an attachment if it exceeds 20,000 characters)

Reviewer #1: (No Response)

7. PLOS authors have the option to publish the peer review history of their article (what does this mean?). If published, this will include your full peer review and any attached files.

If you choose “no”, your identity will remain <gwmw class="ginger-module-highlighter-mistake-type-3" id="gwmw-15768757130591122006919">anonymous but</gwmw> your review may still be made public.

Reviewer #1: No

<gdiv></gdiv>

---

## [Editor Report · Acceptance letter]

30 Dec 2019

PONE-D-19-31014R1 

Health-related quality of life among patients with type 2 diabetes mellitus in Eastern Province, Saudi Arabia: a cross-sectional study 

Dear Dr. Joseph:

I am pleased to inform you that your manuscript has been deemed suitable for publication in PLOS ONE. Congratulations! Your manuscript is now with our production department. 

With kind regards,

on behalf of

Professor Manal S. Fawzy 

Academic Editor

PLOS ONE